# Seeking Pleasure or Meaning? The Different Impacts of Hedonic and Eudaimonic Tourism Happiness on Tourists’ Life Satisfaction

**DOI:** 10.3390/ijerph19031162

**Published:** 2022-01-20

**Authors:** Seolwoo Park, Dongkyun Ahn

**Affiliations:** 1Department of Business, Jeju National University, Jeju 63243, Korea; swpark@jejunu.ac.kr; 2School of Business, Yonsei University, Seoul 03722, Korea

**Keywords:** tourism experiences, hedonic tourism happiness, eudaimonic tourism happiness, life satisfaction

## Abstract

Although hedonic tourism happiness and eudaimonic tourism happiness coexist in tourism experiences, extant research has primarily approached them and their impact on tourists’ life satisfaction separately. Therefore, the purpose of this research is to investigate the impact on life satisfaction of the two types of happiness tourists experience in various activities they encounter in tourist venues and their asymmetric effects. A survey was conducted among tourists who had tourism experiences within a year (October 2018 to September 2019) either abroad or Jeju island, and 736 responses were used in the analysis. Results from structural equation modeling analysis show that most of the hypotheses were supported. Our findings demonstrate that pleasure and detachment experience positively affect hedonic tourism happiness, while personal meaning and self-reflection experiences positively affect eudaimonic tourism happiness. Theoretical and managerial implications are discussed.

## 1. Introduction

Due to recent studies indicating that experiences are more effective in enhancing consumers’ happiness than material goods, researchers are increasingly interested in travel experiences. In particular, extensive research has been conducted on travel consumption, a representative type of experience that increases travelers’ happiness, and its mechanism, as well as variables that slow down the pace of hedonic adaptation to the tourism experience [1,2]. One of the 2019 Travel Trend Keywords released by the Korea Tourism Organization (which analyzed 2.64 million social big-data cases over the past three years) is recreational activities, which indicates that a variety of things to enjoy and experience at tourist venues is an important factor in choosing a travel destination.

Previous studies of the relationship between travel experience and tourists’ happiness have demonstrated that generally, positive tourism experiences increase tourists’ happiness, while disappointing activity experiences at travel destinations have a negative impact [3]. Moreover, Brajša-Žganec et al. [4] indicated that tourists’ happiness was reduced when the experiential activities—including visiting exhibitions and going shopping—did not reflect the age, gender, and interests of tourists. Thus, the key purposes of present-day travel marketing are not simply to increase the number of travelers but to build long-term relationships with tourists, thereby increasing their happiness by managing their experiences at the travel destinations. As a result, considering tourists’ experiences is becoming an increasingly important issue in tourism and marketing [5].

Previous studies related to the happiness experienced by tourists have mainly focused on the hedonic aspects of happiness, which are defined by the frequency and intensity of positive emotional experiences [6]. However, recently, as researchers’ interest in tourist happiness has increased, they have attempted to outline travel happiness in more detail. One of the notable studies on this issue, Nawijn and Filep [7], has divided tourists’ happiness into hedonic and eudaimonic. According to these authors, eudaimonic tourist happiness is defined as travel experience-related happiness that gives tourists not only positive emotions but also meaning and a sense of achievement. However, relatively little effort has been made to identify the integrated impact of tourists’ hedonic and eudaimonic happiness on their overall life satisfaction.

Therefore, in this study, we aim to present an integrated model that covers the effect of tourism activities’ experiential characteristics on two types of tourist happiness (hedonic and eudaimonic) and their impact on overall life satisfaction, focusing on the importance of activities experienced during travel. More specifically, we examine the asymmetric effects of hedonic tourism happiness and eudaimonic tourism happiness on tourists’ life satisfaction, based on an asymmetry study of the effects of existing consumer satisfaction determinants.

The specific objectives of our study are as follows. First, drawing on Waterman et al. [8] eudaimonic identity theory, we try to conceptually divide tourism happiness as experienced by tourists into hedonic and eudaimonic tourism happiness. Second, we attempt to comprehensively verify the impact of tourism activities’ characteristics on the two types of tourism happiness and tourists’ overall life satisfaction. Finally, we seek to confirm the asymmetric effects of the two types of tourism happiness on the overall life satisfaction of tourists. We end with a discussion of the theoretical and practical implications of our findings.

## 2. Literature

### 2.1. Tourist Happiness

#### 2.1.1. Definition and Two Types of Happiness

Happiness experienced by individuals can be defined as a combination of positive emotions and life satisfaction [9]. Happiness has been referred to as a broad concept, a comprehensive term that encompasses not only hedonic characteristics [10,11] but also well-being, quality of life, and subjective well-being resulting from experiencing positive emotions, the fulfillment of various needs, and the achievement of goals [12]. Furthermore, happiness affects life satisfaction at different levels of conceptualization. Therefore, the types of happiness can be divided into hedonic enjoyment and eudaimonia. Hedonic enjoyment involves the immediate joy and satisfaction that individuals get from experience, while eudaimonia consists of individuals experiencing self-expressiveness through activities, which means that the experience gives them meaning and self-definition, and they realize their potential [8,13]. Telfer [14] found that a certain level of effort, flow experience, and self-realization were required to experience eudaimonia; therefore, eudaimonia is not a necessary condition for hedonic enjoyment but a sufficient condition, as shown in Figure 1. In other words, not all experiences that can bring about hedonic enjoyment necessarily reach eudaimonia, which is associated with self-maturation and self-development [8]. Moreover, eudaimonia can be expected to give greater meaning and satisfaction to individuals over the long term by denoting self-expressiveness and the possibility of self-realization.

#### 2.1.2. Tourism Experience Characteristics and Tourists’ Happiness

If we apply the two types of happiness mentioned above to the tourism context, tourism can be divided into two categories: travel with both hedonic enjoyment and eudaimonia and travel with only hedonic enjoyment. While travel researchers who draw on positive psychology have primarily focused on the emotional and hedonic aspects of tourists’ happiness (e.g., pleasure and fun through travel), consumer well-being researchers who take a more cognitive view have concentrated on meeting multi-dimensional needs through travel and self-associated motivations. Sirgy et al. [15] demonstrated that when tourists experienced the gratification of various levels of desires through travel, they felt satisfaction in various domains, and these different dimensions of satisfaction led to positive overall life satisfaction through a positive spillover effect. Bosnjak et al. [16] also confirmed that satisfying intrinsic motivations through self-expressiveness is an important factor in determining tourists’ life satisfaction.

These results are in line with qualitative research, which indicates that travel is helpful for tourists who express their self-image and self-development through post travel experience analysis [17]. Studies using self-determination theory have shown that the fulfillment of intrinsic motivation is more effective than external motivation in increasing the subjective well-being of tourists [18]. Finally, Stebbins [19,20,21] empirically separated leisure activities into serious leisure and casual leisure. According to the author, while casual leisure refers to acts that seek immediate self-satisfaction and pleasure, serious leisure is related to the benefits of self-realization, self-expression, and self-satisfaction. In other words, serious leisure can be defined as a process through which people gain experience and knowledge by engaging in leisure activities and acquiring special skills. Taken together, if prior findings are applied to the two types of happiness, it can be seen that causal leisure is in line with hedonic enjoyment and serious leisure is in line with eudaimonia.

Filep and Deery [1] defined tourists’ happiness as a state in which tourists experience feelings—including positive emotion, engagement, and meaning—that are gained before, during, and after the journey, indicating affective and cognitive harmonies or the combination of external and intrinsic dimensions. Going a step further, Seligman and Flourish [22] presented five factors that make up a traveler’s happiness, based on the automatic happiness theory (PERMA: positive emotion, engagement, relationships, meaning, and achievement). According to these authors, the travel experiences that can contribute to self-realization through a variety of activities have a more lasting effect on tourists’ happiness than simply enjoyable trips.

### 2.2. Tourist Activities

Previous studies on tourism have shown that the most important factor for tourists in the process of considering a destination is the overall image of the travel destination, which also has a significant impact on their happiness [23,24]. However, Oppewal et al. [25] found that the poor image of a destination can be counteracted by engaging tourism experiences through field experiments in Australia. Moreover, once tourists have had a travel experience, they decide to revisit depending on their experience with the tourist activities [26,27]. Thus, it is necessary to investigate experience in tourist activities specifically because these can be very important in the tourism field. Accordingly, we attempt to separate tourist activities at a travel destination into pursuing pleasure and novelty experiences by applying the aforementioned categories of hedonic enjoyment and eudaimonia.

Pursuing pleasure experiences among tourist activities include sightseeing, relaxing in nature, and communicating with locals [28]. In contrast, seeking novelty experiences among tourist activities consist of unique cultural experiences at the destination and special experiences that can only be had at that venue [29]. In a similar vein, Crouch et al. [30] divided the types of travel activities into seven categories: nature experience, self-enhancement, new experiences and knowledge, health and exercise, relaxation and escape, adventure and excitement, family, friends, and fun. They also demonstrated that tourists determine the type of activities they will pursue in the future according to past positive experiences of travel or demographic factors. In other words, tourists seek diversity in travel destinations but remain relatively consistent in the type of tourism experiences and activities, regardless of destination. However, the type of tourist activities pursued at travel destinations changes with age. Older people prefer static activities related to art, history, and nature, while younger people tend to be more engaged in active programs, such as entertainment, festivals, and events [30].

#### Tourists Activities Characteristics and Tourists’ Happiness

In a study of happiness resulting from travel activities, Chen et al. [31] suggested that travel contributes to improving tourists’ quality of life through four types of tourism activity experiences: relaxation, detachment, control, and mastery. Other studies have shown that positive affect, self-expressiveness, and self-realization experiences through travel activities enhance tourists’ quality of life and happiness [15,16,32]. More recently, Knobloch et al. [33] indicated that eudaimonia experiences, such as self-realization activities at travel destinations, have more lasting influences on tourists than pleasure activity experiences. Likewise, Choe et al. [34] found that sharing travel experiences by posting on social media increases tourists’ overall life satisfaction (Table 1).

In addition, studies on the effect of itinerary characteristics on travelers’ happiness showed that travel during holidays increases tourists’ happiness [36] and their pre trip happiness compared to travel during weekdays [37,38]. However, a trip on holidays did not have a significant impact on post travel happiness [39], although if a holiday trip is rest oriented, the post trip happiness of tourists increases [37].

These empirical studies, focused on tourists’ actual travel experiences, have demonstrated that tourists’ travel experiences and activities have a positive effect on their happiness and quality of life.

## 3. Method

### 3.1. Research Model

This study aimed to verify how the empirical characteristics of travel affect two types of traveler happiness—hedonic and eudaimonia—and how they collectively affect travelers’ overall life satisfaction. It also tried to discern the asymmetry of the effect of the two types of happiness on travelers’ life satisfaction. The research model is presented in Figure 2.

### 3.2. Setting Up Hypothesis

Psychologists define hedonic experiences as human behaviors seeking pleasure or avoiding pain [40]. According to previous studies on tourism, hedonic travel experiences are largely divided into ‘experience of pure enjoyment (e.g., visiting a famous theme park at a tourist destination)’ [41], ‘experience of stepping out of the monotonous and daily routines’ [42] (p. 213), and ‘experience of physical and mental recovery away from the rigor of daily life’ [43]. In addition, Waterman [13] demonstrated that activities that provide hedonic pleasure increase hedonic happiness. Furthermore, most studies on the happiness of early travelers assumed that pleasure from traveling and liberation from daily life would lead to an increase in hedonic happiness. The first set of hypotheses established in this study is as follows.

**Hypothesis** **1:**
*The hedonic characteristics of tour activities will increase travelers’ hedonic travel happiness.*


**Hypothesis** **2:**
*The more travelers experience pleasure through tour activities, the higher their hedonic travel happiness.*


**Hypothesis** **3:**
*The more travelers experience detachment through tour activities, the higher their hedonic travel happiness.*


Travelers are not only motivated to acquire hedonic outcomes through travels but are also intrinsically motivated to satisfy their self-related desires (e.g., the realization of the ideal self-image). On the one hand, travelers seek to have hedonic experiences, such as pleasure and escape from reality, through their travels. They also address the value of their lives by performing meaningful and valuable tour activities (e.g., a pilgrimage, etc.), seeking to achieve their goals (e.g., honing their sports skills), and going on eudaimonic travels (e.g., expanding their knowledge of the culture and history in tourist destinations) [44]. Travelers may mature or show personal growth through such eudaimonic travel activities [13]. Therefore, it is expected that the higher the relevance of the tour activities to the discovery of personal meaning—such as the achievement of goals and self-reflection—the greater the feeling of happiness from eudaimonic travels will be [45,46]. Consequently, the second set of hypotheses can be established as follows.

**Hypothesis** **4:**
*The characteristics of eudaimonia in tour activities will increase travelers’ eudaimonic travel happiness.*


**Hypothesis** **5:**
*The more travelers discover personal meaning through tour activities, the greater their eudaimonic travel happiness.*


**Hypothesis** **6:**
*The more travelers have experiences of self-reflection through tour activities, the greater their eudaimonic travel happiness.*


According to the bottom-up spillover theory, satisfaction with various dimensions of life contributes to overall life satisfaction through a series of upward transfer processes [35]. Therefore, experiencing two kinds of happiness through travel will contribute to the attainment of overall life satisfaction. In addition, Stebbins [20] demonstrated that both daily leisure activities with hedonic tendencies and serious activities with eudaimonic tendencies have a significant effect on an individual’s overall quality of life. In addition, Huta and Ryan [47] empirically demonstrated that travelers could experience an improvement in their quality of life when hedonic and eudaimonic travels are properly harmonized [33]. Therefore, it is expected that happiness from hedonic and eudaimonic travels will increase travelers’ overall life satisfaction. Accordingly, Hypothesis 7 was established as follows.

**Hypothesis** **7:**
*Travelers’ happiness from a type of travel will increase their overall life satisfaction.*


**Hypothesis** **8:**
*Travelers’ happiness from hedonic travels will increase their overall life satisfaction.*


**Hypothesis** **9:**
*Travelers’ happiness from eudaimonic travels will increase their overall life satisfaction.*


The next set of hypotheses addresses the asymmetry of the effects of the two kinds of travel happiness on travelers’ life satisfaction. According to the consumption experience analysis based on Herzberg’s two-factor theory, factors of consumption experiences can be classified into hygiene factors and motivating factors, depending on the pattern of the effects on post purchase evaluation. The hygiene factor should be satisfied; however, once needs are satisfied, the hygiene factor is a factor for which the effect is diminished (the negative asymmetry effect), while the motivating factor is a factor that continues to increase in importance even after the basic needs are met (the positive asymmetry effect) [48].

Under self-determination theory, it has been demonstrated that the satisfaction of intrinsic motivation has a more positive effect on the long-term subjective well-being of travelers than the satisfaction of extrinsic motivation does [18]. In other words, it is expected that eudaimonia will continue to affect travelers as a result of its deep connection to the self, even if it does not produce immediate results. Therefore, the effect of happiness from eudaimonic travels on the quality of travelers’ life is expected to have a positive asymmetry effect, in that it continues to increase even after a certain level has been met. However, happiness from hedonic travels results from an immediate outcome of one’s travels, which is likely to disappear quickly. In other words, the effect of happiness from hedonic travels on the quality of travelers’ life has the characteristic of a hygiene factor, in that it diminishes after a specific level has been met and is expected to have a negative asymmetry effect. In line with previous discussions, the fourth hypothesis was established as follows.

**Hypothesis** **10:**
*The two types of travel happiness travelers experience will have an asymmetric effect on their overall life satisfaction.*


**Hypothesis** **11:**
*Happiness from hedonic travels will have a negative asymmetric effect on travelers’ overall life satisfaction.*


**Hypothesis** **12:**
*Happiness from eudaimonic travels will have a positive asymmetric effect on travelers’ overall life satisfaction.*


### 3.3. Data Collection and Questionnaires

This study selected male and female travelers aged 30–70 years who had traveled overseas or to Jeju Island in South Korea between October 2018 and September 2019. This was to ensure the least involvement of tour activities in analyzing their effects on the life satisfaction of adult respondents who had purchasing power. The consumer panel of Hankook Research, one of the research centers, was utilized for the survey, and the research was conducted for approximately one week in October 2019. In general, in determining statistical power, in the case of structural equations, it is recognized that it is desirable to have a sample size of 200 or more [49]. From a total of 800 responses collected, 736 were used in the actual analysis (with unreliable responses excluded).

The demographic characteristics of samples are presented in Table 2. Of the whole valid sample, the proportion of women was 49.94%, and that of men was 50.1%, while 25.3%, 25.4%, 23.6%, and 25.7% of respondents were in their 30s, 40s, 50s, and 60s, respectively, indicating even distribution across gender and age. In the case of respondents’ occupations, office workers accounted for the highest percentage of the total at 39.5%, followed by professionals at 11.5%, homemakers at 9.8%, teachers at 8.3%, service and sales jobs at 7.9%, private operators at 7.1%, and retirees at 6%.

Respondents were asked to recall one of the most memorable trips they had taken in the past year, to briefly describe the activities at the tourist destination, and to respond to several questions about their trip (e.g., I went on a self-guided tour to Hanoi, Vietnam, with my family for four days and three nights during the last summer vacation. At that time, I visited several museums and galleries. I not only got massages at famous massage shops, but I also tasted a variety of traditional Vietnamese dishes.) The average duration of the trips was 5.24 days. Of all tourist destinations, 39.5% were in Jeju Island, South Korea, 25% were in northeast Asia, 21.1% were in southeast Asia, 8.2% were in Europe/Africa, 3.3% were in the U.S./Hawaii, and 2.4% were in the South Pacific. The average number of companions on the trips was 4.37, including the respondents themselves. Approximately 72% went on self-guided tours, and the total average traveling expense per capita was about KRW 1.1 million. This study used questionnaire items that had been validated in previous studies to verify the research hypotheses; the operational definitions of major research concepts are shown in Table 3. The section about pleasure, detachment, personal meaning, and self-reflection consisted of three questions, referring to Lengieza et al. [50] and Waterman [13]. Hedonic tourism happiness section consisted of 4 questions referring to Waterman et al. [8], and eudaimonic tourism happiness consisted of 6 questions referring to Dienner [9]. Specific questionnaire items are presented in Appendix A.

### 3.4. Results

#### 3.4.1. Reliability and Validity Verification

Amos 22.0 was used for the analysis in this study. The questionnaire items were refined through the Cronbach’s alpha and confirmatory factor analysis, the latter conducted to secure the reliability and validity of whole questionnaire items. The variables to be used were determined according to the criteria of factor loadings (reference value: 0.5–0.95) [51]. The value of Cronbach’s alpha ranged from 0.859–0.944 (reference value: acceptable for ≥0.6 and reliable for ≥0.7), and that of composite reliability range from 0.804–0.937, showing the internal consistency of the indicator is secured [51]. The analysis results reveal that the factor loadings (λ) connecting the measurement items and the corresponding constructs are all significant, meaning that the convergence validity is secured (Table 4). Finally, regarding the discriminant validity, in comparisons of the square root of the AVE of each of the two potential factors to be evaluated, if the square roots of the two AVEs are higher than those of the correlation, it can be defined as having discriminant validity. This confirms that the discriminant validity of the corresponding constructs used in this study is secured (Table 5).

#### 3.4.2. Test of Common Method Bias

Given that the data were collected using a self-report survey, there is a possibility of common method bias [52]. In order to confirm the existence of common method bias, three comparative models (M1, M2, and M3) were verified according to the guidelines of Cote and Buckley [53]. M1 is a method-only model (χ^2^(252) = 6183.673, *p* < 0.001) in which all constituent variables measured are included in one factor; M2 is a trait-only model (χ^2^(231) = 833.726, *p* < 0.001) in which each of the measurement items is loaded in a trait factor; and M3 is a method-and-trait model (χ^2^(213) = 605.820, *p* < 0.001) in which the measurement method and concept are considered at the same time (Table 6).

The comparison of the three models shows that M2 and M3 had much better fitness than M1, and the fitness of M3—which simultaneously considered trait factors, measurement methods, and concepts—was statistically significantly better than that of M2, although this difference was much smaller than the difference between M3 and M1 (Δχ^2^(18) = 227.906, *p* < 0.001). This proves that the method used in this study did not have a major effect on the study results; therefore, it can be concluded that common method bias did not occur in this study.

#### 3.4.3. Hypothesis Testing

The analysis results of the structural equation confirm that the fitness of the study model is appropriate in general (χ^2^(243) = 1668.201, *p* < 0.01; CFI = 0.915; GFI = 0.877; NFI = 0.902; RMSEA = 0.089). The analysis result shows that the experience of pleasure has a significant effect on hedonic travel happiness, thereby supporting Hypothesis 2 (Hypothesis 2: path coefficient = 0.566, *p* < 0.05). The analysis result also shows that detachment from daily life has a significant effect on hedonic travel happiness, thereby supporting Hypothesis 3 (Hypothesis 3: path coefficient = 0.277, *p* < 0.05). Hypothesis 5 indicates that the higher the level of personal meaning discovery through travel destination activities, the higher the eudaimonia tourism happiness. The analysis demonstrates that Hypothesis 5 was adopted (Hypothesis 5: path coefficient = 0.209, *p* < 0.05). The analysis also indicates self-reflection affects eudaimonic tourism happiness significantly, thereby supporting Hypothesis 6: path coefficient = 0.517, *p* < 0.05). The analysis results show that both hedonic travel happiness and eudaimonic travel happiness have a significant effect on travelers’ overall life satisfaction, thereby supporting Hypothesis 8 and Hypothesis 9 (hypothesis 8: path coefficient = 0.108, *p* < 0.05; Hypothesis 9: path coefficient = 0.487, *p* < 0.05) (Table 7, Figure 3).

Hypothesis 10 concerns the asymmetry of the effects of happiness from hedonic and eudaimonic travels on travelers’ overall life satisfaction. In other words, it was hypothesized that the effect of eudaimonic travel happiness on life satisfaction has a positive asymmetry that continues to increase even after a certain level has been met (Hypothesis 11). In contrast, the effect of hedonic travel happiness has a negative asymmetry that decreases after a certain level has been met (Hypothesis 12). Dummy variables were additionally created for hedonic happiness and eudaimonic happiness according to the analysis method in previous studies on the asymmetry of the effects (1 for responses higher than the average value, 0 for other responses) [48]. Respondents’ gender and age were used as control variables.

Life satisfaction = β0 + β1·happiness from eudaimonic travels + β2·dummy for happiness from eudaimonic travels + β3·happiness from hedonic travels + β4·dummy for happiness from hedonic travels + ε1

The result of the multiple regression analysis shows that the dummy variables for eudaimonic travel happiness have a significant effect on life satisfaction at the 95% confidence level. It also shows that the value of the dummy variable for hedonic travel happiness is negative but not statistically significant. This indicates that happiness from eudaimonic travels is an improvement factor with a positive asymmetry effect on life satisfaction, but happiness from hedonic travels has a symmetrical effect; therefore, Hypothesis 10 is partially supported (Table 8).

## 4. Conclusions

This study categorized the happiness that travelers experience into two kinds of happiness, one each from hedonic and eudaimonic travels, and conducted an integrated verification of the effects of various activities that travelers experience in tourist destinations on the two kinds of happiness, as well as the effects of such happiness on their overall life satisfaction. The study also verified the asymmetry of the effects of happiness from hedonic and eudaimonic travels on travelers’ overall life satisfaction. The analysis results of a survey with representative consumer panels show that experiencing pleasure from tour activities and experiencing detachment from daily life have a significant effect on happiness from hedonic travels. The discovery of personal meaning and self-reflection have a significant effect on happiness from eudaimonic travels. In addition, the results show that both hedonic and eudaimonic travel happiness have a significant effect on travelers’ life satisfaction. They also show that happiness from eudaimonic travels has the characteristic of an improvement factor, with a positive asymmetry effect on travelers’ life satisfaction, but happiness from hedonic travels has a linear effect. This is meaningful because the obtained results are consistent with the contents explored in previous studies.

The implications of the theoretical and practical aspects of this study are described here. First, this study has shown that the characteristics of various tour activities—in contrast to the approach based on the overall image of travel destinations—contribute to an improvement in happiness from the two kinds of travels, thereby increasing travelers’ overall life satisfaction. As pointed out in previous studies, this can be said to be meaningful, as it shows that the development of specialized activities can be an alternative in overcoming the weak image of tourist destinations and that various activities in tourist destinations contribute substantially to travelers’ life satisfaction.

Second, according to data recently released by TripAdvisor, the world’s largest travel website, the travel trends of tourists continue to change [54]. Specifically, in the past, travelers simply focused on eating food and enjoying themselves in tourist destinations, but in recent years, they have focused on exploring new experiences in tourist destinations, consuming valuable things, and promoting self-reflection through their travels. It can be said, in other words, that the tendency of travelers to pursue eudaimonia is increasing. This study can contribute to finding ways to maximize travelers’ overall life satisfaction through travel happiness by comprehensively incorporating the two types of travel (which have previously been covered individually) into one model.

Lastly, this study has practical implications for industry stakeholders, such as those who plan travel products in travel agencies. According to the results of this study, happiness from hedonic travels has a particular effect on travelers’ life satisfaction, while happiness from eudaimonic travels has a positive asymmetry effect on their life satisfaction. This means that eudaimonia can be an important differentiator in designing travel experiences. Therefore, when developing travel products, travel agencies should identify the needs travelers have, depending on their types and travel purposes, to actively reflect such needs in their products. This will not only increase travelers’ satisfaction with travel products but also maintain the relationship between travel agencies and travelers for a more extended period.

This study’s limitations and future research directions are as follows. First, according to socioemotional selectivity theory, individuals perceive upcoming days in a limited or expansive way according to their current position in their entire life, which is known to affect their goals and motivations [55,56]. Individuals with a limited view not only show more present-oriented aspects but also place more value on emotional goals. In contrast, those with an extended view not only show more future-oriented aspects but also place more value on future-oriented goals (e.g., learning new knowledge). This means that tour activities may have completely different meanings according to an individual’s view of time. Therefore, future studies need to consider a series of processes in which tour activities may contribute to respondents’ life satisfaction by considering their age and view of time.

Second, travelers’ life satisfaction was used as the dependent variable of this study. However, future studies should derive sufficient measures to obtain better marketing performances for travel destinations by using a variety of attitudinal and behavioral variables, such as satisfaction with travel destinations, revisit intention, and word-of-mouth intention for travel destinations.

Third, this study verified the effect of travel experience traits on travelers’ life satisfaction through a survey; future studies should carry out longitudinal studies on how satisfying one’s self-related needs through travel experiences has a long-term effect on travelers’ awareness of life.

Despite the limitations mentioned above, this study confirmed the effects of tour activities on travelers’ two kinds of happiness and their life satisfaction. It also verified the asymmetry of the effects of happiness from two types of travels on travelers’ life satisfaction. Finally, the results of this study are expected to inform further studies that seek to increase the well-being of travelers.

## Figures and Tables

**Figure 1 ijerph-19-01162-f001:**
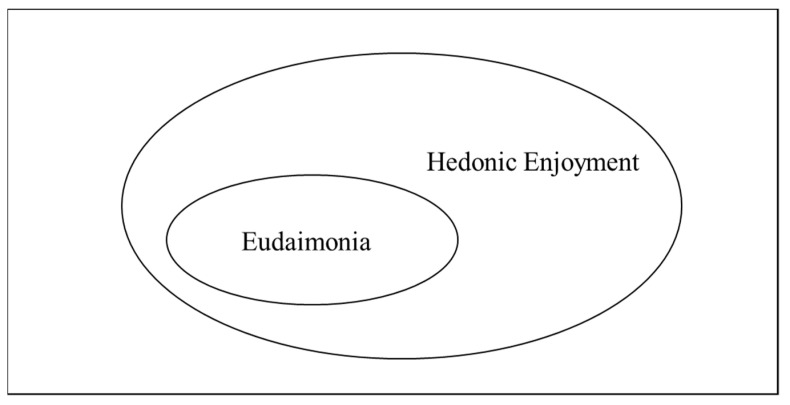
Two Types of Happiness.

**Figure 2 ijerph-19-01162-f002:**
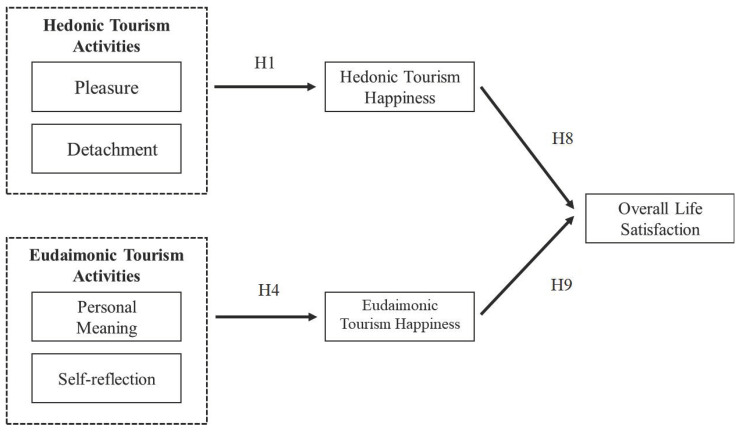
Research Model.

**Figure 3 ijerph-19-01162-f003:**
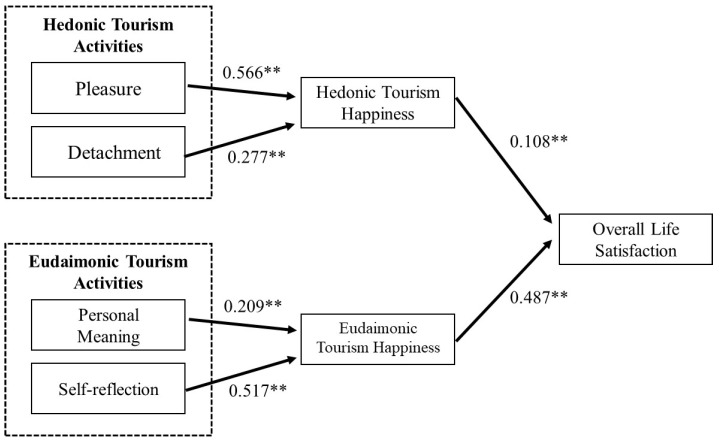
Path analysis results.

**Table 1 ijerph-19-01162-t001:** Summary Review of Travel Experience and Travelers’ Happiness Research.

Authors	Major Findings
Chen et al. [31]	Travel increases the life satisfaction of travelers through four types of recovery experiences (relaxation, detachment, control, and mastery experience).
Bosnjak et al. [16]	The self-expressiveness and pleasure of travelers through sports leisure travel positively affect travelers’ happiness.
Hosany et al. [32]	Emotional experience is a key variable in the satisfaction of travel destinations and the happiness of travelers.
Sirgy et al. [35]	Achieving goals through leisure travel improves the quality of life of leisure travelers.
Knobloch et al. [33]	The memorable trip is highly correlated to eudaimonia related to self-realization.
Choe et al. [34]	In addition to the hedonic experience, attention is needed for eudaimonia, which relates to self-realization and personal meaning.

**Table 2 ijerph-19-01162-t002:** Sample Characteristics.

Age	Frequency (Percentage)	Occupation		Annual Travel Period	
1981–1990 (30–39)	186 (25.3%)	Private operator	7.1%	Less than 5 days	33.0%
1971–1980 (40–49)	187 (25.4%)	Service and sales	7.9%	5~9 days	36.2%
1961–1970 (50–59)	174 (23.6%)	Manufacturing	6.3%	More than 10 days	30.8%
1951–1960 (60–69)	189 (25.7%)	R&D	3.7%		
		Office worker	39.5%		
**Gender**	**Frequency (Percentage)**	Professional	11.5%		
Female	367 (49.9%)	Teacher	8.3%		
Male	369 (50.1%)	Housewife	9.8%		
		Retiree	6.0%		

**Table 3 ijerph-19-01162-t003:** Operational Definitions of Major Concepts.

Construct	Operational Definition	Source
Pleasure	Pleasure generated by the travel experience	Lengieza et al. [50]
Detachment	Avoidance from reality caused by the travel experiences	
Personal Meaning	Inner maturity gained from travel experience	
Self-reflection	Self-reflection from travel experience	Waterman [13]
Hedonic Tourism Happiness	Pleasant happiness in the dimension of travel life felt by travelers	Waterman et al. [8]
Eudaimonic Tourism Happiness	Eudaimonia on the dimension of travel life felt by travelers	Dienner [9]

**Table 4 ijerph-19-01162-t004:** Confirmatory Factor Analysis.

Construct	Items	Loading	*t*-Value	Cronbach’s Alpha	AVE	Composite Reliability (CR)
Pleasure	EHEP1	0.868	14.32	0.914	0.751	0.901
	EHEP2	0.885	13.428			
	EHEP3	0.898	12.515			
Detachment	EHEA1	0.802	14.726	0.859	0.577	0.804
	EHEA2	0.842	13.098			
	EHEA3	0.815	14.263			
Personal Meaning	EEEP1	0.852	15.415	0.909	0.686	0.908
	EEEP2	0.904	12.773			
	EEEP3	0.877	14.418			
Self-reflection	EEES1	0.862	15.022	0.909	0.681	0.853
	EEES2	0.886	13.884			
	EEES3	0.884	13.989			
Hedonic Tourism Happiness	HTH1	0.914	14.077	0.944	0.833	0.937
	HTH2	0.935	12.147			
	HTH3	0.917	13.829			
Eudaimonic Tourism Happiness	ETH1	0.861	16.235	0.934	0.671	0.911
	ETH2	0.881	15.606			
	ETH3	0.804	17.309			
	ETH5	0.887	15.391			
	ETH6	0.875	15.802			
Overall Life Satisfaction	SWB1	0.852	14.947	0.918	0.672	0.891
	SWB2	0.901	12.265			
	SWB3	0.87	14.148			
	SWB4	0.815	16.048			

χ^2^(*p* value) = 833.726(0.000), df = 231, CFI = 0.964, GFI = 0.915, NFI = 0.951, RMSEA = 0.060.

**Table 5 ijerph-19-01162-t005:** Correlation Matrix Between the Constructs (Phi matrix).

	a	b	c	d	e	f	g
Pleasure (a)	0.751						
Detachment (b)	0.750	0.577					
Personal Meaning (c)	0.593	0.672	0.686				
Self-reflection (d)	0.576	0.665	0.906	0.681			
Hedonic Tourism Happiness (e)	0.766	0.695	0.506	0.540	0.833		
Eudaimonic Tourism Happiness (f)	0.700	0.716	0.710	0.739	0.826	0.671	
Overall Life Satisfaction (g)	0.442	0.437	0.473	0.464	0.491	0.587	0.672

Correlations matrix among latent variables (i.e., Phi correlations) with AVE on the diagonal.

**Table 6 ijerph-19-01162-t006:** Test of Common Method Bias.

Model	Goodness-of-Fit Statistics	Results
M1: Method-only model	χ^2^(252) = 6183.673, *p* < 0.001, GFI = 0.472, CFI = 0.647, NFI = 0.638, RMSEA = 0.179	
M2: Trait-only model	χ^2^(231) = 833.726, *p* < 0.001, GFI = 0.915, CFI = 0.964, NFI = 0.951, RMSEA = 0.060	
M3: Method-and-trait model	χ^2^(213) = 605.820, *p* < 0.001, GFI = 0.936, CFI = 0.977, NFI = 0.965, RMSEA = 0.050	Δχ^2^(18) = 227.906, *p* < 0.001

**Table 7 ijerph-19-01162-t007:** Hypothesis test.

Hypotheses	Structural Relationships	Path Coefficient	*t*-Value
H2	Pleasure → Hedonic Tourism Happiness	0.566 **	11.525
H3	Detachment → Hedonic Tourism Happiness	0.277 **	5.748
H5	Personal Meaning → Eudaimonic Tourism Happiness	0.209 **	2.441
H6	Self-reflection → Eudaimonic Tourism Happiness	0.517 **	6.138
H8	Hedonic Tourism Happiness → Overall Life Satisfaction	0.108 **	3.233
H9	Eudaimonic Tourism Happiness → Overall Life Satisfaction	0.487 **	13.829

χ^2^(*p*-value) = 1668.201(0.00), df = 243; CFI = 0.915; GFI = 0.877; NFI = 0.902; RMSEA = 0.089. ** Significant at 95% CI.

**Table 8 ijerph-19-01162-t008:** Results of Asymmetry Effect.

DV	Overall Life Satisfaction	Estimate	*t*-Value
IV	Hedonic Tourism Happiness	0.151 **	2.295
	Hedonic Tourism Happiness Dummy	−0.024	−0.422
	Eudaimonic Tourism Happiness	0.372 **	5.579
	Eudaimonic Tourism Happiness Dummy	0.105	1.871

Note. Gender and age effects were not statistically significant. ** Significant at 95% CI.

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
