# Peer review of "Seeking Pleasure or Meaning? The Different Impacts of Hedonic and Eudaimonic Tourism Happiness on Tourists’ Life Satisfaction"

_ijerph, 2022, doi:10.3390/ijerph19031162_

Round 1

Reviewer 1 Report

While this is a very interesting paper, the authors should revise some issues:

- in the abstract, the authors should also explain the method they used.

- while the introduction is well elaborated, recent references should be added to the section.

- please add a short paragraph to present the structure of section 2 and section 3.

- the authors should conceptualize the understanding of the tourist experience, and add more recent research about the relationships between happiness and tourism. The theoretical review should be more critical than descriptive.

- please identify the hypothesis 3 before the sub hypotheses.

- the authors say that ‘this study selected male and female travelers aged 30–70 years who had traveled 267 overseas or to Jeju Island in South Korea’. The rationale of the sample and the method should be explained.

- the authors should present the results as a separate chapter and expand the dialogue with previous literature.

- the section 5 should be renamed as conclusions.

Author Response

Response to Reviewer 1 Comments

Point 1: in the abstract, the authors should also explain the method they used.

Response 1: According to the reviewer's suggestion, we added an explanation of the methodology to the abstract.

“A survey was conducted for tourists who had tourism experiences within a year (October 2018 to September 2019) either to abroad or Jeju island, and 736 responses were used in the analysis. Results from structural equation modeling analysis show that most of the hypotheses were supported.”

Point 2: while the introduction is well elaborated, recent references should be added to the section.

Response 2: In the introduction, we revealed the motivation for the start of the study through the latest data in 2019. Please check.

“One of the 2019 Travel Trend Keywords released by the Korea Tourism Organization (which analyzed 2.64 million social big-data cases over the past three years) is recreation-al activities, which indicates that a variety of things to enjoy and experience at tourist venues is an important factor in choosing a travel destination.”

Point 3: please add a short paragraph to present the structure of section 2 and section 3.

Response 3: We revealed the purpose and flow of the study in the introduction. Please check.

“Therefore, in this study, we aim to present an integrated model that covers the effect of tourism activities’ experiential characteristics on two types of tourist happiness (he-donic tourism happiness and eudaimonic tourism happiness) and their overall life satis-faction, focusing on the importance of activities experienced during travel. More specifi-cally, we try to examine the asymmetric effects of hedonic tourism happiness and eudai-monic tourism happiness on tourists’ life satisfaction, based on an asymmetry study of the effects of existing consumer satisfaction determinants.”

Point 4: the authors should conceptualize the understanding of the tourist experience, and add more recent research about the relationships between happiness and tourism. The theoretical review should be more critical than descriptive.

Response 4: Data from this study were collected before COVID-19. The concept of happiness was dealt with in Chapter 2, and we tried to refer to the research of the 2010s as much as possible. Please check.

Point 5: please identify the hypothesis 3 before the sub hypotheses.

Response 5: According to the reviewer's suggestion, we added H3.

“Hypothesis 3: Travelers’ happiness from type of travels will increase their overall life satisfaction.”

Point 6: the authors say that ‘this study selected male and female travelers aged 30–70 years who had traveled 267 overseas or to Jeju Island in South Korea’. The rationale of the sample and the method should be explained.

Response 6: We collected a total of 800 respondents and used 736 data. We explained the validity of sample collection in the part of method. Please check.

“This study selected male and female travelers aged 30–70 years who had traveled overseas or to Jeju Island in South Korea between October 2018 and September 2019 as re-spondents. This was to ensure the least involvement of tour activities in analyzing their effects on the life satisfaction of adult respondents who had purchasing power. The con-sumer panel of Hankook Research, one of the research centers, was utilized for the survey, and the research was conducted for approximately one week in October 2019. Of a total of 800 responses collected, 736 were used in the actual analysis (with unreliable responses excluded).”

Point 7: the authors should present the results as a separate chapter and expand the dialogue with previous literature. the section 5 should be renamed as conclusions.

Response 7: According to the reviewer's suggestion, We named Section 5 as a conclusion and divided it.

Thank you for your good suggestion.

Reviewer 2 Report

The manuscript investigates the impact of the hedonic tourism happiness and the eudaimonic tourism happiness on life satisfaction. The paper is very interesting and well-written. Congratulations to the Authors!

Below some comments (minor revisions) to improve its overall quality:

I suggest to shortly describe the case study in introduction and stressing its relevance.

Figure 2 and hypotheses should be moved to section 2. These are theoretical aspects that do not fit into research methods. If Authors believe that in this way section 2 becomes too long, they may create a new specific section (that could be named as “Model and hypotheses setting”) to be placed between literature and materials and methods.

In section 3 Authors should add a short sub-section to describe the study area (context of analysis): a map, some figures about tourism arrivals and, as for the introduction, a description of its relevance could help the reader to contextualize the empirical investigation.

Finally Authors report the questionnaire in the Appendix and this is fine. However, a brief description of the questionnaire in section 3.3 can be helpful.

Sections 4 and 5 are OK.

Author Response

Response to Reviewer 2 Comments

Point 1: I suggest to shortly describe the case study in introduction and stressing its relevance.

Response 1: In the introduction, we revealed the motivation for the start of the study through the case study. Please check.

“One of the 2019 Travel Trend Keywords released by the Korea Tourism Organization (which analyzed 2.64 million social big-data cases over the past three years) is recreation-al activities, which indicates that a variety of things to enjoy and experience at tourist venues is an important factor in choosing a travel destination.”

Point 2: Figure 2 and hypotheses should be moved to section 2. These are theoretical aspects that do not fit into research methods. If Authors believe that in this way section 2 becomes too long, they may create a new specific section (that could be named as “Model and hypotheses setting”) to be placed between literature and materials and methods.

Response 2: In many studies, the hypothesis proposal part is introduced after presenting a research model in the methodology section. Please understand it.

Point 3: In section 3 Authors should add a short sub-section to describe the study area (context of analysis): a map, some figures about tourism arrivals and, as for the introduction, a description of its relevance could help the reader to contextualize the empirical investigation.

Response 3: We explained the relevance through the description of the sample in the methodology part. Please understand it.

“This study selected male and female travelers aged 30–70 years who had traveled overseas or to Jeju Island in South Korea between October 2018 and September 2019 as re-spondents. This was to ensure the least involvement of tour activities in analyzing their effects on the life satisfaction of adult respondents who had purchasing power. The con-sumer panel of Hankook Research, one of the research centers, was utilized for the survey, and the research was conducted for approximately one week in October 2019. Of a total of 800 responses collected, 736 were used in the actual analysis (with unreliable responses excluded).”

Point 4: Finally Authors report the questionnaire in the Appendix and this is fine. However, a brief description of the questionnaire in section 3.3 can be helpful.

Response 4: All questions were shown in the appendix, but we briefly introduced them in section 3.3  according to the reviewer's suggestion.

“The questions about pleasure, detachment, personal meaning, and self-reflection consist-ed of three questions, referring to are Lengieza et al. [49] and Waterman [13]. Hedonic tourism happiness consisted of 4 questions referring to Waterman et al. [8], and eudai-monic tourism happiness consisted of 6 questions referring to Dienner [9].”

Thank you for your good suggestion.

Round 2

Reviewer 1 Report

While the authors have provided a cover letter with the revisions, the new version is not submitted with tracked changes. Also, the authors have not revised the majority of the issues mentioned in the previous report. I send again the issues which require further attention by the authors, all of them from the previous report:

- while the introduction is well elaborated, recent academic and industry references should be added to the section.

- the authors have not added a short paragraph to present the structure of section 2 and section 3.

- the authors should conceptualize the understanding of the tourist experience, and add more relevant research about the relationships between happiness and tourism. The theoretical review should be more critical than descriptive.

- also, the authors say that ‘this study selected male and female travelers aged 30–70 years who had traveled 267 overseas or to Jeju Island in South Korea’. The rationale of the sample and the method should be justified with proper references.

- the authors should expand the dialogue between results and previous literature.

Author Response

Response to Reviewer 1 Comments

Point 1: while the introduction is well elaborated, recent academic and industry references should be added to the section.

Response 1: In the introduction, we revealed the motivation for the start of the study through the latest data in 2019. Please check.

“One of the 2019 Travel Trend Keywords released by the Korea Tourism Organization (which analyzed 2.64 million social big-data cases over the past three years) is recreation-al activities, which indicates that a variety of things to enjoy and experience at tourist venues is an important factor in choosing a travel destination.”

Furthermore, we added Kwon & Lee (2020)’s study.

Point 2: the authors have not added a short paragraph to present the structure of section 2 and section 3.

Response 2:  I do not understand the reviewer's suggestion. We revealed the purpose and flow of the study in the introduction.

“Therefore, in this study, we aim to present an integrated model that covers the effect of tourism activities’ experiential characteristics on two types of tourist happiness (he-donic tourism happiness and eudaimonic tourism happiness) and their overall life satis-faction, focusing on the importance of activities experienced during travel. More specifi-cally, we try to examine the asymmetric effects of hedonic tourism happiness and eudai-monic tourism happiness on tourists’ life satisfaction, based on an asymmetry study of the effects of existing consumer satisfaction determinants.”

In addition, we wrote the paper by referring to the structure and flow of other papers published in IJERPH.

(e.g. Ko, Y.; Lee, H.; Hyun, S.S. Airline Cabin Crew Team System’s Positive Evaluation Factors and Their Impact on Personal Health and Team Potency. Int. J. Environ. Res. Public Health 2021, 18, 10480.

Kang, J.; Song, Y. A Study on the Relationship between Mental Well-Being and Cultural Tourism Guides Based on the Interview Methodology. Int. J. Environ. Res. Public Health 2021, 18, 13054.

Byun, H.-J.; Lee, B.-C.; Kim, D.; Park, K.-H. Market Segmentation by Motivations of Urban Forest Users and Differences in Perceived Effects. Int. J. Environ. Res. Public Health 2022, 19, 114.

Point 3: the authors should conceptualize the understanding of the tourist experience, and add more relevant research about the relationships between happiness and tourism. The theoretical review should be more critical than descriptive.

Response 3: In the literature review, we first divided and conceptualized traveler's happiness into hedonic and eudaimonic based on previous studies. Next, the characteristics of traveler's experience were classified, and the relationship with traveler's happiness was explained earlier. In addition, the relationship between the characteristics of traveler's behavior(activity) and traveler's happiness that can be embodied in the traveler's experience was explained step by step. Please understand it.

Point 4:  also, the authors say that ‘this study selected male and female travelers aged 30–70 years who had traveled 267 overseas or to Jeju Island in South Korea’. The rationale of the sample and the method should be justified with proper references.

Response 4: We collected a total of 800 respondents and used 736 data. We explained the validity of sample collection in the part of method. Following reviewer’s suggestion we added sample justification (Kline, 2005).

“This study selected male and female travelers aged 30–70 years who had traveled overseas or to Jeju Island in South Korea between October 2018 and September 2019 as re-spondents. This was to ensure the least involvement of tour activities in analyzing their effects on the life satisfaction of adult respondents who had purchasing power. The con-sumer panel of Hankook Research, one of the research centers, was utilized for the survey, and the research was conducted for approximately one week in October 2019. In general, in determining statistical power, in the case of structural equations, it is recognized that it is desirable to have a sample size of 200 or more [49]. Of a total of 800 responses collected, 736 were used in the actual analysis (with unreliable responses excluded).”

Point 5: the authors should expand the dialogue between results and previous literature.

Response 5: In the first part of the conclusion, we summarized the results of the hypothesis described in previous studies. For the reader's understanding, we once again expressed that the contents explained in previous studies were consistent with the results according to the reviewer's suggestion.

“This study categorized the happiness that travelers experience into two kinds of hap-piness, one each from hedonic and eudaimonic travels, and conducted an integrated veri-fication of the effects of various activities that travelers experience in tourist destinations on the two kinds of happiness, as well as the effects of such happiness on their overall life satisfaction. The study also verified the asymmetry of the effects of happiness from he-donic and eudaimonic travels on travelers’ overall life satisfaction. The analysis results of a survey with representative consumer panels show that experiencing pleasure from tour activities and experiencing detachment from daily life have a significant effect on happi-ness from hedonic travels, while the discovery of personal meaning and self-reflection have a significant effect on happiness from eudaimonic travels. In addition, the results show that both hedonic travel happiness and eudaimonic travel happiness have a signif-icant effect on travelers’ life satisfaction. They also show that happiness from eudaimonic travels has the characteristic of an improvement factor, with a positive asymmetry effect on travelers’ life satisfaction, but happiness from hedonic travels has a linear effect. This is meaningful in that the results consistent with the contents explored in previous studies were obtained.”

Thank you for your good suggestion.
